# Low Anterior Resection Syndrome: What Have We Learned Assessing a Large Population? [note 1]

**DOI:** 10.3390/jcm11164752

**Published:** 2022-08-15

**Authors:** Audrius Dulskas, Povilas Kavaliauskas, Edgaras Kulikauskas, Edgaras Smolskas, Kornelija Pumputiene, Narimantas E. Samalavicius, Joseph W. Nunoo-Mensah

**Affiliations:** 1Department of Abdominal and General Surgery and Oncology, National Cancer Institute, 1 Santariskiu Str., LT-08406 Vilnius, Lithuania; 2Faculty of Health Care, University of Applied Sciences, 45 Didlaukio Str., LT-08303 Vilnius, Lithuania; 3Institute of Clinical Medicine, Faculty of Medicine, Vilnius University, LT-03101 Vilnius, Lithuania; 4Department of Public Health, Institute of Health Sciences, Faculty of Medicine, Vilnius University, LT-03101 Vilnius, Lithuania; 5Utena Hospital, 3 Aukstakalnio Str., LT-28151 Utena, Lithuania; 6Vilnius City Clinical Hospital, 57 Antakalnio Str., LT-10207 Vilnius, Lithuania; 7Baltupiu Family Clinics, 80A Didlaukio Str., LT-08326 Vilnius, Lithuania; 8Klaipeda University Hospital, Liepojos Str., LT-92288 Klaipeda, Lithuania; 9King’s College Hospital NHS Foundation Trust, London SE5 9RS, UK; 10Cleveland Clinic London, London SW1X 7HY, UK

**Keywords:** anterior resection, bowel dysfunction, cohort study, LARS score, low anterior resection syndrome score, rectal cancer

## Abstract

Our goal was to assess the rate of symptoms commonly included in LARS score in a large general population. The study was based on a population-based design. We disseminated LARS scores through community online platforms and general practitioners throughout Lithuania. We received 8183 responses to the questionnaire. There were 142 (1.74%) participants who were excluded for lack of information. There were 6100 (75.9%) females and 1941 (24.1%) males. After adjusting for sex and age, male participants had a significant average score of 18.4 (SD ± 10.35) and female 20.3 (SD ± 9.74) *p* < 0.001. There were 36.4% of participants who had minor LARS symptoms, and 14.2% who had major LARS symptoms. Overall, major LARS-related symptoms were significantly related to previous operations: 863 participants in the operated group (71.7%), and 340 in the non-operated group (28.3%; p0.001). In 51–75-year-old patients, major LARS was significantly more prevalent with 22.7% (*p* < 0.001) and increasing with age, with a higher incidence of females after the age of 75. After excluding colorectal and perineal procedures, the results of multivariate logistic regression analysis indicated the use of neurological drugs and gynaecological operations were independent risk factors for major LARS–odd ratio of 1.6 (*p* = 0.018, SI 1.2–2.1) and 1.28 (*p* = 0.018, SI 1.07–1.53), respectively. The symptoms included in the LARS score are common in the general population, and there is a variety of factors that influence this, including previous surgeries, age, sex, comorbidities, and medication. These factors should be considered when interpreting the LARS score following low anterior resection and when considering treatment options preoperatively.

## 1. Introduction

As the survival rate of rectal cancer has increased in recent years, the issue of low anterior resection syndrome (LARS) has gained more attention and discussion among colorectal surgeons since this syndrome adversely impacts patient functionality and quality of life following a successful anterior resection [1,2,3,4]. Even in the presence of good sphincter and nerve sparing techniques, approximately 90% of patients who undergo sphincter-preserving low anterior resection surgery will experience a change in their bowel habits ranging from increased bowel frequency to faecal incontinence (FI) or evacuatory dysfunction [5]. The wide range of symptoms that may be associated with post-rectal resection and reconstruction has been termed low anterior resection syndrome, whose severity may also reflect a significant decline in the quality of life (QOL) of patients [6]. As recently as 2012, a Danish research group standardized and validated a questionnaire to quantify the situation for each individual patient and to formulate therapeutic approaches aimed at improving the condition [7]. The questionnaire has been translated and validated into many languages, including Lithuanian [8,9,10].

Recently, three studies, assessing almost 4000 ‘healthy’ individuals, were published [11,12,13]. According to the authors, the prevalence of major LARS varied from 9.6–18.8% based on age and sex. Jull et al. suggested that this prevalence should be considered normative for the assessment of LARS following a low anterior resection [11]. Moreover, a baseline measurement of LARS at the time of the cancer diagnosis is often a poor estimate of a patient’s true function before he/she was affected by the cancer. Normative data from the general population can serve as a baseline in studies assessing rectal cancer patients. This data reflect the level of bowel function disturbances in a background population similar in other parameters, but unaffected by rectal cancer.

The purpose of our study was to conduct a large survey of Lithuanians using the LARS questionnaire and to evaluate possible risk factors for symptoms that may also occur in LARS. 

## 2. Patients and Methods

Between July and December 2019, a Lithuanian translation of the LARS questionnaire was distributed throughout Lithuania using community online platforms and general practitioners [9]. For the promotion of the questionnaire, we used social information groups on Facebook, e-mails to patient organizations, and our social contacts (face-to-face interviews). The surveys could not be duplicated by filling in two or more from the same IP address. Out of approximately 20,000 people who were contacted to participate in this study, 8183 (40.9%) responded by completing the questionnaire. Each participant completed the survey voluntarily and anonymously. In total, 40.9% of the questionnaires were returned; however, 142 respondents (1.74%) were excluded due to improperly completed forms. As with the standard LARS questionnaire, the Lithuanian LARS questionnaire consisted of five questions [9]. The questionnaire score ranges from 0 to 42 points, with respondents being classified as having no LARS (0 to 20 points), minor LARS (21 to 29 points), or major LARS (30 to 42 points). Each question has three or four different responses. Additionally, participants were asked to provide basic demographic information. These included age, sex, and medications (none, antihypertensive drugs, diabetic medications, neurological medications, etc.). We also collected information regarding past surgical history. This included no operations or procedures that involve laparotomy for abdominal organs (stomach, liver, small bowel, liver, pancreas, etc) or a hernia, gynaecological, urological, perineal (haemorrhoids, fissures, fistulas), colorectal (colon and rectal procedures) or at other sites, such as orthopaedic, dermatological, head and neck, etc. Finally, our analysis evaluated healthy populations, which had no previous operations or had declared that they had no underlying diseases and repeated risk factor assessment.

## 3. Study Groups

Our initial analysis considered all the participants’ sex, age, medication use, and history of surgical operations. Subsequently, we adjusted for sex and age, since there was a discrepancy between the percentage of males and females in our study population. Approximately 90% of colorectal cancers occur in people aged 51 to 75 years old, and we concentrated on the analysis of this group of patients [14]. In addition, we performed age adjusted analysis. The statistical analysis was performed using IBM SPSS v23 (IBM Corp., Armonk, NY, USA) and Microsoft Excel 2010 (Microsoft Corp., Redmond, WA, USA). Descriptive statistics were used to analyse the basic characteristics. For comparing groups or means, the Chi-square and Student T-tests were used. A *p*-value < 0.05 was considered statistically significant. A multivariate and univariate logistic regression analysis was used to evaluate the odds ratio (OR) and the associated 95% confidence interval (CI).

## 4. Results

The basic demographic characteristics of our sampled population are presented in Table 1. Most responders were females. Almost two thirds of responders were in the 31–75 age group. More than half the population declared that they do not use any medications; almost 40% had no previous operations. Male participants had an average LARS score of 18.4 (±SD 10.35), while females had a score of 20.1 (±SD 9.83) which was statistically significant (*p* < 0.001). After adjustment for sex and age, we found similar statistically significant results: the score for men was 18.4 (±SD 10.35) compared to females 20.3 (±SD 9.74) which was also significant (*p* < 0.001). Minor LARS accounted for 36.4% of our study population after adjusting for age and sex, and major LARS accounted for 14.2%. Additionally, after adjusting for age and sex, major LARS was significantly more common in the female population by a difference of 9.4% and 1.2 times (Table 2). Overall, major LARS is associated with 1179 total operations, and including those with multiple overall previous operations, there were 863 participants in the operated group (71.7%) and 340 participants (28.3%) in the non-operated group (*p* < 0.001)—Table 2.

The prevalence of major LARS increases with age and with a higher female predisposition towards the age of 75 (Figure 1). Major LARS was significantly more prevalent in the 51–75 age patient group with 22.7% (*p* < 0.001)—see Table 2 and Figure 1. Following the age of 75, males continue to experience a proportional increase in major LARS, whereas this trend was reversed for females (Figure 1). There was no significant difference in age distribution between the sexes.

Table 3 represents statistically significant risk factors for minor LARS. Multivariate logistic regression of the entire study population showed that male sex was an independent factor for lower risk for minor LARS, as well as age groups less than 30 years and 30–50 years (Table 4). A multivariate logistic regression analysis of the whole study population showed that major LARS is less common in patients who do not take medications. Furthermore, independent risk factors for major LARS included neurological drugs and gynaecological and colorectal operations (Table 4). For the 51–75-year-old group, which has a high incidence of colorectal cancer, univariate logistic regression analysis demonstrated sex, the general use of medications, neurological drugs, other medications, previous procedures, abdominal operations, and gynaecological surgeries as risk factors for major LARS (Table 4). Multivariate logistic regression analysis showed that use of neurological drugs was an independent risk factor for major LARS for 51–75-year-old group with a significance of 1.79 OR and 2.59, respectively, (Table 5).

Furthermore, when we excluded all respondents who had previous procedures or who took medications and then calculated the LARS score again, 2491 responders were left, including 782 males (31.4%) and 1709 females (68.6%). Of these, 1140 (45.8%) responders were younger than 30 years, 1233 (49.5%) were in age 31 to 50, and only 118 (4.7%) responders were older than 50. For young males (i.e., <30 years old), the average LARS score was 17.23 (SD ± 10.05), whereas the average score for females was 18.72 (SD ± 9.75) (*p* < 0.001). Among the healthy group (those without previous procedures and without taking any medications), 1368 respondents (54.9%) did not have LARS, 877 (35.2%) had minor LARS, and 246 (9.9%) had major LARS.

## 5. Discussion

In our study, we found a reasonable number of the “heathy” population suffering from LARS-like symptoms.

Several older population-based studies have demonstrated that up to 15% of the general population have faecal incontinence (FI) [15,16]. Moreover, a recent systematic review of the prevalence of faecal incontinence found rates ranging from 1.4 to 19.5% from 30 studies that were analysed out of 4840 articles searched [17]. The variation was explained by differences in the data collection method and two factors within the definitions of FI: type of stool and frequency of episodes of FI. When the prevalence estimate of functional FI was studied in the five studies that used the Rome II criteria, the combined functional FI prevalence was 5.9% (95% CI 5.6–6.3%) [17]. The first two questions of the LARS questionnaire evaluate faecal incontinence, and the rest of the questions evaluate rectal capacity, compliance, and evacuative function. Therefore, the problem here is that the LARS questionnaire is not specific to LARS.

We observed that, after adjusting for age and sex, minor LARS symptoms accounted for 36.4% and major LARS symptoms for 14.2% of our study population. Major LARS symptoms were more prevalent in patients 51–75 years of age with 22.7%. Major LARS symptoms increased with increasing age to 75 and were more prevalent in females. These results are in accordance with other studies, which have reported LARS scores ranging between 9.6 and 18.8% of the general population [11,12,13]. Juul et al. published normative data on LARS scores among a population of 1875 Danes aged 20 to 89 years in 2019 [11]. Factors associated with major LARS were physical illness and female sex. Among females between 50 and 79 years, 19% reported major LARS, while 10% of males in the same age group reported major LARS. In another study of a Dutch reference population, the prevalence of major LARS was 15% [12]. Meanwhile, in a healthy Dutch cohort of 1259 individuals, minor and major LARS were detected in 24.3% and 12.2%, respectively [13].

In our univariate analysis, we found that female sex, older age, previous surgery, and usage of medication were risk factors for LARS symptoms. Based on our multivariate analysis, only colorectal surgery and neurological drugs were significant risk factors for major LARS symptoms, which is in contrast to previous studies. In assessing the prevalence of LARS symptoms in a healthy Danish population, the authors of this publication found that only people with diabetes were at greater risk for LARS [13]. Juul et al. found that in the Danish general population, major LARS was associated with female sex and physical disease, while age was not associated statistically significantly with major LARS [11]. In addition, a study from the Netherlands demonstrated that only female sex was a risk factor for LARS [12]. As a result of the systemic review discussed above regarding FI, it was found that older age, diabetes mellitus, urinary incontinence, frequent and loose stools, and multiple chronic illnesses were all risk factors for FI [17]. These risk factors also overlap with some of the risk factors we and other researchers discovered in our study of LARS in healthy individuals. The explanation for the poor bowel function observed in women may be related to the shorter anal canal leading to urgency. Furthermore, during childbirth, women tend to experience some degree of injury to the sphincter complex. In older patients, the sphincter and pelvic floor muscles become weaker, leading to faecal incontinence and other bowel movement disorders 

Interestingly, 51% of our Lithuanian population group suffered from LARS without having previously undergone a rectal resection. It is even more striking that LARS symptoms (minor or even major LARS) can occur in a population without having used medication or having previous surgery (41.1%). The LARS score questionnaire appears to have a high sensitivity for symptoms of bowel dysfunction, but a limited specificity for LARS. Ribas et al. questioned the usefulness of a single questionnaire in a clinical setting and conducted a prospective study on 70 patients who had undergone a curative anterior resection for rectal cancer [18]. Due to the complexity of the LARS, the authors concluded that the LARS score may overestimate the impact on quality of life for some patients and underestimate the impact of severe evacuatory dysfunction. In their opinion, a single LARS questionnaire may not be sufficient to assess bowel function, and a full clinical evaluation and additional questionnaires may be necessary [18]. Since LARS is a multifactorial problem because of rectal surgery, we should also approach it from a multifaceted perspective. Nevertheless, the LARS score continues to be the most effective tool for assessing LARS longitudinally and evaluating treatment effectiveness. Further, we believe the LARS score should be tested in individuals with other bowel diseases/dysfunction, such as inflammatory bowel disease. Afterwards, we should be able to obtain more nominative information regarding the questionnaire. We recommend that all patients complete a LARS score preoperatively while we are presently evaluating the effectiveness of this method as part of an international clinic effectiveness study conducted by the International Society of University Colon and Rectal Surgeons. Tumour volume and stenosing of the lumen may mimic LARS symptoms. In addition, some bowel function symptoms associated with the LARS syndrome (such as FI, clustering, and urgency) may also occur in the general population. In our study, we also found that the general population has a high prevalence of these symptoms. It is imperative that surgeons recognize and take these additional risk factors into account as they evaluate their surgical options in those who ultimately require LAR for cancer. Moreover, LARS was developed based upon a constellation of symptoms, rather than a consensus definition among patients and clinicians. Authors in a study by Chen et al. assessed cancer specialists and patients with rectal cancer and concluded that specialists lack a comprehensive understanding of which bowel dysfunction symptoms are important to the patient, nor how these symptoms affect quality of life (QOL) [19]. A study by Van der Heijden et al. demonstrated the importance of patient-generated recommendations [20]. It is possible that these are the reasons why the score is sensitive but not specific.

Recently, the Delphi consensus on LARS description was published. To be considered as having LARS, a patient must undergo an anterior resection (sphincter-preserving resection) and experience at least one of eight symptoms that result in at least one of eight consequences [21]. A distinct advantage of the Delphi approach is that, unlike most patient-reported outcome measures that are initially developed by experts in the field of clinical research, who then consult patient populations, the Delphi definition of LARS actively involves all major stakeholders, especially patients, early on in the development process to ensure that the resulting tool fits its purpose. Our opinion is that this definition of LARS should be standard in any department performing colorectal surgery, but the LARS score might nonetheless be a useful tool to collect pre- and postoperative data. In patients who require a LAR, the ideal is to collect a preoperative LARS score, and then adjust the postoperative LARS score accordingly. It should also be noted that there is a preoperative assessment tool for identifying possible postoperative LARS—Pre-Operative LARS score (POLARS) [22]. As with other original studies, it had limited accuracy in predicting the risk and severity of LARS [23].

We are aware of the potential limitations of our study. First, we have only investigated a few of the possible comorbidities and factors that may influence LARS symptoms. Secondly, our study was restricted to one country’s population, so we were unable to assess whether the demographics in our sample are representative of those of the entire population. Thirdly, there is always the possibility of selection bias affecting response rates. For example, individuals with existing health concerns or bowel issues may be more inclined to respond to the survey than otherwise healthy individuals who may ignore it. Despite this, we believe that a high number of included individuals reduces the risk to a minimum. There were more female responders to our questionnaire, which may indicate the discomfort or embarrassment experienced by males in addressing this issue. Moreover, most of the questionnaires were filled out online without the assistance of a specialist to explain any question. Lastly, we are familiar with other risk factors for bowel dysfunction, such as a history of childbirth trauma or radiation to the pelvis. Despite this, our study is the largest single cohort study and includes more respondents than previous combined studies.

## 6. Conclusions

The symptoms commonly attributed to LARS are actually caused by other conditions and diseases, as well as being influenced by age, sex, comorbidities, and medication use. When assessing bowel function following low anterior resection, all these factors should be considered perioperatively. Moreover, a LARS questionary should be performed preoperatively (POLARS) and postoperatively.

## Figures and Tables

**Figure 1 jcm-11-04752-f001:**
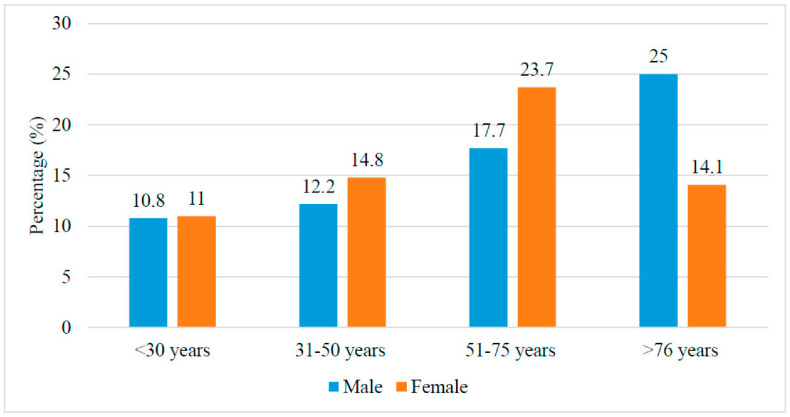
Major low anterior resection syndrome (LARS) prevalence comparison according to age and sex (*p* < 0.05).

**Table 1 jcm-11-04752-t001:** Demographics of population included in the study.

Variable	Count
Sex	
Male (%)	1941 (24.1%)
Female (%)	6100 (75.9%)
Age groups	
<30 years (%)	2349 (29.2%)
31–50 years (%)	3985 (49.6%)
51–75 years (%)	1592 (19.8%)
>76 years (%)	115 (1.4%)
Use of medications:	
None (%)	5443 (67.7%)
Antihypertensive (%)	762 (9.5%)
Antidiabetic (%)	151 (1.9%)
Nervous system affecting (%)	424 (5.3%)
Others (%)	1678 (20.9%)
Previous operations (some patients had multiple operations)	
None (%)	3104 (38.6%)
Abdominal (%)	1575 (19.6%)
Gynaecological (%)	1832 (22.8%)
Urological (%)	248 (3.1%)
Perineal (%)	174 (2.2%)
Colorectal (%)	151 (1.9%)
Other (%)	2351 (29.1%)

**Table 2 jcm-11-04752-t002:** Low Anterior Resection Syndrome score (LARS) score correlation to various factors.

Variable	No LARS (%)	Minor LARS (%)	Major LARS (%)	*p* Value
Sex				
Female	2896 (47.5%)	2251 (36.9%)	953 (15.6%)	<0.001 *
Male	1033 (53.2%)	658 (33.9%)	250 (12.9%)	
Sex (after adjusting for age and sex)				
Female	885 (45.6%)	754 (38.4%)	302 (16%)	<0.001 *
Male	1033 (53.2%)	658 (33.9%)	250 (12.9%)	
Age (years)				
<30	1175 (50%)	917 (39%)	257 (11%)	<0.001 *
31–50	1918 (48.1%)	1503 (37.7%)	564 (14.2%)	
51–75	770 (48.4%)	461 (29.0%)	361 (22.7%)	
>76	66 (57.4%)	28 (24.3%)	21 (18.3%)	
Medication use	2800 (51.4%)	1986 (36.5%)	657 (12.1%)	
No medication	364 (47.8%)	245 (32.2%)	153 (20.1%)	
Antihypertensive drugs	66 (43.7%)	43 (28.5%)	42 (27.8%)	<0.001 *
Antidiabetic drugs	151 (35.6%)	155 (36.6%)	118 (27.8%)	
Neurological drugs	705 (42.0%)	614 (36.6%)	359 (29.8%)	
Previous operations				
No operation	1612 (53.0%)	1092 (35.9%)	340 (11.2%)	
Abdominal	704 (44.7%)	592 (37.6%)	279 (23.2%)	<0.001 **
Gynaecological	790 (43.1%)	674 (36.8%)	368 (20.1%)	<0.001 **
Urological	115 (46.4%)	79 (31.9%)	54 (21.8%)	0.008 **
Perineal	71 (40.8%)	69 (39.7%)	34 (19.5%)	0.066 **
Colorectal	54 (35.8%)	45 (29.8%)	52 (34.4%)	<0.001 **
Other	1107 (47.3%)	841 (35.9%)	394 (16.8%)	0.009 **

*—Compared within the groups; **—compared to no operations.

**Table 3 jcm-11-04752-t003:** Univariate analysis for risk factors for minor low anterior resection syndrome (LARS) score of whole population.

Factor	Odds Ratio	*p*	95% CI
Male sex Female sex	0.692 1	0.029	0.49–0.96
General use of medication	1.57	0.001	1.2–2.04
Nervous system affecting drugs	2.1	0.001	1.49–2.95
Other medication	1.46	0.002	1.15–1.85
Previous operations	1.73	0.005	1.18–2.55
Abdominal operations	1.3	0.04	1.01–1.69
Gynecological operations	1.38	0.008	1.09–1.75
Colorectal operations	2.62	0.001	1.73–3.96

**Table 4 jcm-11-04752-t004:** Multivariate analysis of risk factors for minor low anterior resection syndrome (LARS) score of whole population.

Factor	Odds Ratio	*p*	95% CI
Male sex	0.86	0.005	0.77–0.95
Female sex	1		
Age (years)			
<30	1.95	0.003	1.26–3.01
30–50	1.85	0.005	1.2–2.84

**Table 5 jcm-11-04752-t005:** Multivariate analysis of risk factors for major low anterior resection syndrome (LARS) score of whole population (**A**) and in 50 to 75 years old patients’ group (**B**).

A			
Factor	Odds Ratio	*p*	95% CI
No medication use	0.61	0.001	0.53–0.69
Nervous system affecting drugs	1.53	0.001	1.2–1.94
Gynaecological operations	1.41	0.001	1.22–1.64
Colorectal operations	2.55	0.001	1.8–3.6
**B**			
**Factor**	**Odds Ration**	** *p* **	**95% CI**
Nervous system affecting drugs	1.79	0.002	1.25–2.57
Colorectal operations	2.59	<0.001	1.69–3.96

## Data Availability

Data is available from the corresponding authors upon reasonable request.

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
