# Peer review of "Low Anterior Resection Syndrome: What Have We Learned Assessing a Large Population? [Author-notes fn1-jcm-11-04752]"

_jcm, 2022, doi:10.3390/jcm11164752_

Round 1

Reviewer 1 Report

The article „low anterior resection syndrome. What have we learned assessing a large population?” investigates the prevalence of symptoms described in the LARS score in a large lithuanian population.

Approximately 20.000 randomly chosen persons were addressed through online platforms and general practitioners and asked to fill out the LARS questionnaire as well as to give informations about their medical history including medications and surgical procedures. Approximately 40% of people were responders. The results were stratified for the LARS score (no LARS, minor and major LARS), gender, use of medication and previous surgical procedures.

The results includes the demographic analysis of the cohort including 76% female and 24% male people. The mean LARS score was 18 for male and 20 for female people. Female people had a frequency of 54.7% and male people of 45.3%  of major LARS in the total age adjusted group This difference was statistically significant higher in female people. There was an univariate and multivariate analysis for minor and major LARS showing that the use of nervous system affecting drugs as well as gynecological and colorectal operations were associated with significantly higher Odds ratio of LARS (Table 4). Univariate and multivariate analysis was also performed for minor LARS showing that male gender is associated with lower risk for LARS.

Omitting all people with medication and surgical procedures resulting an a “healthy” cohort, 9.9% showed signs of major LARS.

The authors conclude that the LARS score is widely detectable in a large “healthy” people cohort and that the symptoms described in the LARS score are non-specific and should be used cautiously.  

Major concerns:  the presentation of the results is somehow “unsharp”. For example, in Table 2, it is unclear which data are significantly different by the statistical analysis. There are no legends helping to understand it. Furthermore, in Figure 1, statistical analysis of the data is significantly different between which groups( line 145).  However which groups in the figure are compared resulting in a p of 0.000 ?. There is no asterix in the figure.

Furthermore, in “patients and methods” the authors first describe that the surgical history of the patient cohort was reported and analyzed but then they stated that “our analysis excluded participants who had previously undergone colorectal or perineal surgery” (line 106). Where these patients only excluded in the last analysis “healthy cohort” (line 168-174) ?

Sentences such as “individuals who do not take medications have a lower risk of developing major LARS” (line 149 and line 222) suggest that there is a causal relationship between the use and non-use of medication and LARS. However, the results do not allow to conclude a causal relationship. These statements should be rephrased.

Minor concerns:

1.      Line 112 “mg 90%” should be approximately 90% ?

2.      Line 145 (p<0.000) – is that correct ?

3.      Was there a difference in age distribution between female and male people in the cohort ?

Author Response

Dear Reviewer,

Thank you for your letter and constructive comments concerning our manuscript entitled “Low anterior resection syndrome. What have we learned assessing a large population?”. The paper was revised substantially. Following changes have been made. They are as follows:

Major concerns:  

the presentation of the results is somehow “unsharp”. For example, in Table 2, it is unclear which data are significantly different by the statistical analysis. There are no legends helping to understand it.

Edited to make it clearer

Furthermore, in Figure 1, statistical analysis of the data is significantly different between which groups (line 145).  However which groups in the figure are compared resulting in a p of 0.000?. There is no asterix in the figure.

Supplemented to make the speech clearer

Furthermore, in “patients and methods” the authors first describe that the surgical history of the patient cohort was reported and analyzed but then they stated that “our analysis excluded participants who had previously undergone colorectal or perineal surgery” (line 106). Where these patients only excluded in the last analysis “healthy cohort” (line 168-174) ?

Changed sentence in line 106 as per suggestion. 

Sentences such as “individuals who do not take medications have a lower risk of developing major LARS” (line 149 and line 222) suggest that there is a causal relationship between the use and non-use of medication and LARS. However, the results do not allow to conclude a causal relationship. These statements should be rephrased.

Changed as per suggestion.

Minor concerns:

  1. Line 112 “mg 90%” should be approximately 90% ?

Changed as per suggestion.

  1. Line 145 (p<0.000) – is that correct ?

Corrected to p<0.05

  1. Was there a difference in age distribution between female and male people in the cohort ?

No there was no difference in age distribution between the sexes. The sentence was added to the RESULTS section.  

The manuscript improved a lot. Thank you once again.

Sincerely

Reviewer 2 Report

Thank you for giving me the chance for reviewing the manuscript “Low anterior resection syndrome. What have we learned 2 assessing a large population?” by Dulskas et al.

They could clearly show risk factors for LARS in a big population independent of rectal cancer surgery which is very interesting but not really new. Prior publication I have some major concerns:

1.     Introduction:

The question of this study is weak and could be explained in a more precise way. Only the last sentence of the introduction leads to your question. Maybe you can improve this. Moreover, the first part of the discussion section is more an introduction.

2.     Please change gender into sex in the entire manuscript. In addition, if you cite somebody it should be ….et al.

3.     Methods:

P5 L112 spelling error (mg).

Please add the head office of Microsoft

P5 L113: you made an analysis of the entire cohort and an age adjusted analysis in addition. You did not focus only on the cohort of 51-74 years old people. Please correct.

4.     Results

a)     You should start with an overview on the entire cohort (how many participants etc.), then you should present patients characteristics. The presentation of the results is very confusing.

b)    The next important information is table 2. In this table you show the LARS in relation to age, gender etc. But this is not a univariate analysis. In addition, percentages should be presented other way round. In my opinion the percentages should be in relation of the number of male (53%) and female (35%) participants not in relation to the entire cohort.

c)     Then you should present the univariate results.

d)    In the text you describe that male sex was an independent factor for a lower risk for LARS. This fact is right and good described in previous literature. But in your tables (univariate and multivariate analysis) you always described male sex as a risk factor. Please revise your data carefully. If your analysis in the tables is right, male sex is a RISK factor for LARS, respectively.

e)     After finishing with the multivariate analysis, you switch again to the univariate analysis. This is confusing and should be corrected.

5.     Discussion:

The first part of the discussion is an introduction. However, the discussion contains information which is not really in context with your study. A more precise focus and discussion of your results is eligible.

a)     Sense of this sentence is hard to understand. Please correct. “Originally this 178 questionnaire was designed to assess bowel function in low anterior resection patients, but it 179 quickly gained wide acceptance and validation for the assessment of LARS.”

b)    L207: “LARS questionary is not specific for LARS”. I think you should delete this sentence. The LARS score was initially defined for patients who underwent a low anterior resection and not for a “healthy” population. To adjust this score to healthy people is a good idea and you are right that the questionary is not very specific but to date this is the only short questionary for patients who undergo a low anterior resection. Moreover, this questionary is at the moment the only good tool to quantify postoperative symptoms. However, if you want to discuss the specificity of LARS you may have to change the name of your questionary because you deleted the patients who underwent a low anterior resection. One important thing is mentioned in your study (this is the most important point): A LARS questionary should be performed preoperatively and postoperatively. This is the key message of your study in my opinion.

c)     L221: female sex or male sex as risk factor for LARS. The interpretation of risk factors in your tables is confusing. Please correct.

d)    The paragraph 268-281 gives no additional information to your study.

e)     As you mentioned in the limitations section, this study contains a huge selection bias. Therefore, the last sentence of the discussion should be revised. It can be doubt that the main conclusion can be generalized globally.  

Language and spelling could be improved

Author Response

Dear Reviewer,

Thank you for your letter and constructive comments concerning our manuscript entitled “Low anterior resection syndrome. What have we learned assessing a large population?”. The paper was revised substantially. Following changes have been made. They are as follows:

Introduction:

The question of this study is weak and could be explained in a more precise way. Only the last sentence of the introduction leads to your question. Maybe you can improve this. Moreover, the first part of the discussion section is more an introduction.

We have removed the first paragraph of the DISCUSSION. Introduction was rewritten to concentrate more on our study

  1. Please change gender into sex in the entire manuscript. In addition, if you cite somebody it should be ….et al.

Gender as replaced to sex all over the text. Citing was corrected as per suggestion

  1. Methods:

P5 L112 spelling error (mg).

Changed as per suggestion. Thank you.

Please add the head office of Microsoft

Inserted as per suggestion.

P5 L113: you made an analysis of the entire cohort and an age adjusted analysis in addition. You did not focus only on the cohort of 51-74 years old people. Please correct.

This is correct – corrected.

  1. Results
  2. a)You should start with an overview on the entire cohort (how many participants etc.), then you should present patients characteristics. The presentation of the results is very confusing.

      Full demographics are shown in Table 1. Added text with overview on the entire cohort.

  1. b)The next important information is table 2. In this table you show the LARS in relation to age, gender etc. But this is not a univariate analysis. In addition, percentages should be presented other way round. In my opinion the percentages should be in relation of the number of male (53%) and female (35%) participants not in relation to the entire cohort. 

       Totally agree. Changed as per suggestion.

  1. c)Then you should present the univariate results.

      Changed as per suggestion.

  1. d)In the text you describe that male sex was an independent factor for a lower risk for LARS. This fact is right and good described in previous literature. But in your tables (univariate and multivariate analysis) you always described male sex as a risk factor. Please revise your data carefully. If your analysis in the tables is right, male sex is a RISK factor for LARS, respectively. – in text we described male gender as a factor for lower risk, not as a RISK factor.

      We added the line in the table with comparison to female gender.

  1. e)After finishing with the multivariate analysis, you switch again to the univariate analysis. This is confusing and should be corrected.

      Changed as per suggestion.

  1. Discussion:

The first part of the discussion is an introduction. However, the discussion contains information which is not really in context with your study. A more precise focus and discussion of your results is eligible.

The paragraph was partially moved to Introduction. Information not related with our study was removed from the Discussion.  

  1. Sense of this sentence is hard to understand. Please correct. “Originally this 178 questionnaire was designed to assess bowel function in low anterior resection patients, but it 179 quickly gained wide acceptance and validation for the assessment of LARS.”

The sentence was removed from the discussion.

  1. L207: “LARS questionary is not specific for LARS”. I think you should delete this sentence. The LARS score was initially defined for patients who underwent a low anterior resection and not for a “healthy” population. To adjust this score to healthy people is a good idea and you are right that the questionary is not very specific but to date this is the only short questionary for patients who undergo a low anterior resection. Moreover, this questionary is at the moment the only good tool to quantify postoperative symptoms. However, if you want to discuss the specificity of LARS you may have to change the name of your questionary because you deleted the patients who underwent a low anterior resection. One important thing is mentioned in your study (this is the most important point): A LARS questionary should be performed preoperatively and postoperatively. This is the key message of your study in my opinion.

Thank you for the suggestion. We totally agree with you point and the key message. This was added to the conclusions. The sentence was deleted.

  1. L221: female sex or male sex as risk factor for LARS. The interpretation of risk factors in your tables is confusing. Please correct.

Supplemented tables to comparison to female gender.

  1. d)The paragraph 268-281 gives no additional information to your study.

      We agree. However, the Delphi consensus made a lot changes in our study, as It was started before the consensus and following it, we had to at least mention it just to support our study. If it looks too away from our study, we obviously can delete it.

  1. e)As you mentioned in the limitations section, this study contains a huge selection bias. Therefore, the last sentence of the discussion should be revised. It can be doubt that the main conclusion can be generalized globally.  

 We have deleted a sentence to prevent confusion. Thank you!

Language and spelling could be improved

Language and spelling actually was revised by native English literature teacher.

The manuscript improved a lot. Thank you once again.

Sincerely

Round 2

Reviewer 1 Report

The paper has improved.

Author Response

Dear reviewer,

Thank you for your valuable comments.

Sincerely

Reviewer 2 Report

Thank you very much for your review!

Some minor comments:

1. Table 2: Percentages should be presented other way round. In my opinion the percentages should not be in relation to the presented groups. Please correct.

2. Spelling error: Line 210 (healthy)

Author Response

Dear Reviewer,

We have made the suggested changes.

Thank you in advance

Sincerely
